WhoseEgg: classification software for invasive carp eggs

Goode Katherine 1 katherine.j.goode@gmail.com
Weber Michael J. 2
Dixon Philip M. 1
1 Department of Statistics, Iowa State University , Ames, Iowa , United States
2 Natural Resource Ecology and Management, Iowa State University , Ames, Iowa , United States
Ward Eric
Electronic publication date: 2023 Feb 27
Publication date: 2023
Volume: 11
Electronic Location ID: e14787
Received 2022 Oct 20; Accepted 2023 Jan 3
Copyright: © 2023 Goode et al.
Copyright year: 2023
Copyright holder: Goode et al.
License: This is an open access article distributed under the terms of the Creative Commons Attribution License, which permits unrestricted use, distribution, reproduction and adaptation in any medium and for any purpose provided that it is properly attributed. For attribution, the original author(s), title, publication source (PeerJ) and either DOI or URL of the article must be cited.
License URL: https://creativecommons.org/licenses/by/4.0/

Keywords: Bigheaded carp, Invasive species, Machine learning, Morphometrics, R Shiny, Random forests, Reproduction

Funding: Iowa Department of Natural Resources 14CRDFBGSCHO-0001 US Fish and Wildlife Services F16AP00791 This work was funded by the Iowa Department of Natural Resources through contract 14CRDFBGSCHO-0001 and by US Fish and Wildlife Services through contract F16AP00791. The funders had no role in study design, data collection and analysis, decision to publish, or preparation of the manuscript.

==============================
The collection of fish eggs is a commonly used technique for monitoring invasive carp. Genetic identification is the most trusted method for identifying fish eggs but is expensive and slow. Recent work suggests random forest models could provide an inexpensive method for identifying invasive carp eggs based on morphometric egg characteristics. While random forests provide accurate predictions, they do not produce a simple formula for obtaining new predictions. Instead, individuals must have knowledge of the R coding language, limiting the individuals who can use the random forests for resource management. We present WhoseEgg: a web-based point-and-click application that allows non-R users to access random forests via a point and click interface to rapidly identify fish eggs with an objective of detecting invasive carp (Bighead, Grass, and Silver Carp) in the Upper Mississippi River basin. This article provides an overview of WhoseEgg, an example application, and future research directions.

Introduction

The collection and identification of fish eggs is a common practice for monitoring invasive aquatic species. By collecting eggs, it is possible to relate environmental conditions to the timing of reproduction, estimate spawning locations, identify potential recruitment bottlenecks between early life stages, and estimate adult spawning biomass (Leggett & Deblois, 1994; Takasuka, Yoneda & Oozeki, 2019; Camacho et al., In press). As a result, egg identification is a useful tool for understanding the reproductive processes of invasive species, which helps to monitor the range of the species and inform management decisions (MICRA, 2017).

Egg collection is one method used to monitor the spread of invasive Grass Carp (Ctenopharyngodon idella), Silver Carp (Hypophthalmichthys molitrix), and Bighead Carp (H. nobilis; Deters, Chapman & McElroy, 2013; Coulter et al., 2016; Embke et al., 2016). Hereafter, Grass Carp, Silver Carp, and Bighead Carp are collectively referred to as “invasive carp”. Invasive carp were introduced to the United States in the 1960s (Freeze & Henderson, 1982; Wittmann et al., 2014) and have spread throughout the Mississippi River basin via natural and anthropogenic means (Chick & Pegg, 2001; Hinterthuer, 2012). Invasive carp alter food webs and fish communities through alterations in nutrient cycling (Collins & Wahl, 2017), reductions in plankton resources, competition with native planktivorous fishes, and reductions in native fish recruitment (Irons et al., 2007; Chick et al., 2020; Tillotson, Weber & Pierce, 2022). Identifying when and where invasive carp are reproducing could inform management efforts seeking to limit their spread into new habitats (e.g., installation of deterrents).

Taxonomic keys are available to identify fish larvae (e.g., Auer, 1982). Fish eggs can also have distinguishing morphological features but are much more difficult to visually identify (Kelso, Kaller & Rutherford, 2012), making the visual identification of invasive carp eggs unreliable (USGS, 2014; Larson et al., 2016). The visual identification of invasive carp eggs is challenging for two reasons. First, morphological features of invasive carp eggs overlap with native species (Chapman, 2006; George & Chapman, 2015; Camacho et al., 2019). Second, invasive carp egg characteristics are not only plastic between their native and invaded regions (Mack et al., 2000; Peterson & Vieglais, 2001; Lenaerts et al., 2015) but can also vary within invaded areas (Lenaerts et al., In press). Currently, genetic identification is the most accurate method for identifying fish eggs (Becker et al., 2015; Coulter et al., 2016; Embke et al., 2016) but it is time intensive and expensive, limiting the number of eggs that can be processed. Consequently, eDNA techniques are being developed to determine if target species are present within a sample (e.g., Fritts et al., 2018), but additional work is needed to identify large numbers of individual fish eggs more easily, quickly, and inexpensively.

Camacho et al. (2019) provides one solution by using random forest machine learning models (Breiman, 2001) to predict the family, genus, and species levels of fish eggs collected in pools 17–20 of the Upper Mississippi River basin during 2014 and 2015. Genetic identifications of eggs were used as response variables in the models (one model for each taxon) with invasive carp treated as one prediction class within each taxonomic level. The predictor variables were 17 egg and environmental characteristics associated with egg collection. Random forests predicted at least 97% of invasive carp eggs correctly at all three taxonomic levels from out-of-bag samples. Goode et al. (In press) validated the models from Camacho et al. (2019) using a set of eggs collected from a third year (2016) across a larger geographic area (pools 14–20). The validation revealed models from Camacho et al. (2019) predicted at least 89% of invasive carp eggs correctly. Additionally, Goode et al. (In press) trained new random forests using the same structure as the models in Camacho et al. (2019) but with all 3 years of egg data. These models returned predictive accuracies for invasive carp between 96% and 98% on the out-of-bag samples. Performance of the models on the validation data suggests random forests can be a useful tool for identifying invasive carp eggs.

Random forests are a relatively new model (Breiman, 2001) that have been applied to a wide range of ecological questions (e.g., Cutler et al., 2007; Evans & Cushman, 2009; Darling et al., 2012). Random forests are desirable because they often produce more accurate model predications compared to more traditional statistical approaches (e.g., logistic regression; Cutler et al., 2007). Yet, a downside of random forests is that the algorithm is too complicated to be written as a predictive equation in a practical form to make predictions for new observations. Instead, a saved version of the model must be accessed directly to obtain predictions. Camacho et al. (2019) and Goode et al. (In press) trained their random forests using the R statistical coding language (R Core Team, 2021). As a result, if an individual is interested in using the random forests from either Camacho et al. (2019) or Goode et al. (In press), it is necessary for the individual to be familiar with the R programming language, limiting the individuals that can access and apply the models for the identification of invasive carp eggs. While random forests could be used as an inexpensive tool to classify invasive carp eggs, there is a need to make the models more broadly available.

We developed the online application of WhoseEgg to allow users unfamiliar with the R programming language to use random forest models to classify invasive carp eggs in the Upper Mississippi River basin. Users can upload their own fish egg characteristics and compute family, genus, and species taxonomic level predictions using random forests based on those from Camacho et al. (2019) and Goode et al. (In press). This article introduces and provides an overview of the capabilities of WhoseEgg. In particular, the article (1) describes how WhoseEgg is accessed and structured, (2) provides details about the training data and random forests used by WhoseEgg, (3) describes the processes for measuring egg characteristic, (4) includes an example demonstrating WhoseEgg, (5) discusses the limitations of WhoseEgg, and (6) suggests directions for future work.

App access and architecture

WhoseEgg is free and available online at https://whoseegg.stat.iastate.edu/. The app is accessible from any device with a web browser but was developed to perform best when used on a laptop or desktop computer. The app was built using R code (R Core Team, 2021) and the R package Shiny (Chang et al., 2021). WhoseEgg is hosted on an R server that allows the app to connect to R to perform the necessary computations. Data uploaded to WhoseEgg will not be saved or redistributed in any manner to protect the privacy of users’ data. The code, random forests, and training data associated with WhoseEgg are available on GitHub (https://github.com/goodekat/WhoseEgg) and in the Iowa State University digital repository (https://doi.org/10.25380/iastate.15046578; version 1.0.0).

WhoseEgg is divided into six pages listed in the top panel of its ‘Home’ page (Fig. 1). The pages are organized so that users begin at the ‘Home’ page and progress left to right through the other pages. The ‘Home’ page contains information to familiarize users with WhoseEgg, including its purpose and instructions. The ‘Home’ page also includes details about the collection locations and species in the training data to help users determine whether the models in WhoseEgg are appropriate for their data.

Figure 1 Homepage of WhoseEgg.

The homepage contains a description of the app and instructions on how to use the app to obtain fish egg taxonomic predictions.

The ‘Data Input’, ‘Predictions’, and ‘Downloads’ pages contain interactive tools that allow users to provide their own data and acquire predictions. Each of these pages is divided into two panels: an instruction panel and a main panel (e.g., Fig. 2). The main panels contain additional information and interactive features to assist users. The flowchart included on the WhoseEgg ‘Home’ page describes the steps to obtain predictions (Fig. 1).

Figure 2 ‘Data Input’ page.

The content in WhoseEgg after the example spreadsheet of fish egg characteristics were uploaded is depicted.

Data Input: The user uploads a spreadsheet with the necessary egg characteristics (Table 1) via the ‘Data Input’ page. The spreadsheet must be an Excel or csv file and formatted appropriately. The ‘Data Input’ page describes the necessary spreadsheet format and provides a downloadable Excel template (included in the Supplemental Material). The template has data validation helpers to further assist users with formatting (Figs. 3A and 3B). Additionally, informative errors and warnings appear in WhoseEgg if the uploaded data are not formatted correctly.

Predictions: The user obtains predictions from the WhoseEgg random forests for the uploaded egg data on the ‘Predictions’ page.

Downloads: The user downloads a spreadsheet from the ‘Downloads’ page containing the uploaded data, additional egg characteristics computed by WhoseEgg, and the random forest predictions.

Table 1 WhoseEgg random forest predictor variables.

The table contains the WhoseEgg random forest predictor variables with definitions and training data means (and standard deviations) or levels (and proportion of eggs per level).

Variable	Definition	Mean (standard deviation) or levels (proportion)	
Compact or diffuse	Whether the embryo is compact or diffuse	Compact (0.87); Diffuse (0.13)	
Conductivity (µ/cm)	Conductivity of the water at the time of collection	462.21 (103.03)	
Deflated membrane	Whether the membrane is deflated or not	Yes (0.59); No (0.41)	
Egg stage	Stage of the egg when collected (based on Kelso & Rutherford (1996))	1 (0.15); 2 (0.01); 3 (0.09); 4 (0.22); 5 (0.10); 6 (0.12);
7 (0.10); 8 (0.08); Broken (<0.01); Diffuse (0.13)	
Embryo diameter average (mm)	Average of four measurements of the embryo diameter	1.36 (0.42)	
Embryo diameter coefficient of variation	Coefficient of variation of four measurements of the embryo diameter	0.1 (0.08)	
Embryo diameter standard deviation (mm)	Standard deviation of four measurements of the embryo diameter	0.14 (0.14)	
Embryo to membrane ratio	Ratio of the embryo diameter average to the membrane diameter average	0.67 (0.2)	
Julian day	Julian day when the egg was collected	167.99 (27.21)	
Larval length (mm)	Length along the midline for all eggs in stages 6–8 (otherwise set to 0)	0.49 (1.12)	
Membrane diameter average (mm)	Average of four measurements of the membrane diameter	2.27 (1.04)	
Membrane diameter coefficient of variation	Coefficient of variation of four measurements of the membrane diameter	0.07 (0.07)	
Membrane diameter standard deviation (mm)	Standard deviation of four measurements of the membrane diameter	0.17 (0.17)	
Month	Month when the egg was collected	5.85 (0.98)	
Pigment presence	Whether there is pigment present on the egg	Yes (0.29); No (0.71)	
Sticky debris	Whether there is debris on the egg	Yes (0.23); No (0.77)	
Temperature (°C)	Temperature of the water when the egg was collected	23.39 (2.95)	

Figure 3 Examples of the validation helpers in the spreadsheet template to assist users correctly format the egg characteristic data.

(A) When a column is selected, a description of the variable and necessary format appears. (B) If an observation is entered incorrectly or falls outside of the range of the WhoseEgg training data, an error/warning appears.

The ‘Help’ and ‘References’ pages are designed to be accessed at any time. The ‘Help’ page contains the details of how to measure the egg characteristics, an overview of random forests, and answers to frequently asked questions. The ‘References’ page lists citations mentioned throughout the app.

Training data and random forests

WhoseEgg uses three random forests to separately predict the family, genus, and species of a fish egg based on egg characteristics. We trained the random forests using a compilation of the training data from Camacho et al. (2019; 734 and 541 fish eggs from 2014 and 2015, respectively) and the validation data from Goode et al. (In press; 703 fish eggs from 2016). The eggs in both studies were sampled from locations in the Upper Mississippi River basin (Fig. 4). The data collection was approved by the Iowa State University Institutional Animal Care and Use Committee Protocol (7-13-7599-I), and the Iowa DNR gave permission for field sampling (SC1037). The data sets contain genetic identifications and egg characteristics. See Camacho et al. (2019) and Goode et al. (In press) for additional details about the egg sampling, subsampling, genetic identification, and egg characteristic measurement procedures. We identified 29 eggs from 2016 with incorrect data entries. We were able to correct 23 of the observations and removed six of the eggs. Thus, the WhoseEgg random forests were trained on a total of 1,972 eggs but provide comparable estimates to the Goode et al. (In press) combined validation models (analysis and results included in the Supplemental Material).

Figure 4 Upper Mississippi River and tributary rivers in Iowa and Illinois, USA where eggs were collected.

The symbols indicate the year(s) of collection: 2014–2015 (plus), 2016 (star), or 2014–2016 (diamond). Map of sampling locations acquired from Goode et al. (In press).

The predictor variables in the WhoseEgg random forests were the same 17 egg and environmental characteristics (Table 1) used in the random forests from Camacho et al. (2019) and Goode et al. (In press). The response variables were the genetically identified family, genus, and species levels of the eggs. For all three taxonomic levels, Grass Carp, Silver Carp, and Bighead Carp were grouped into the category of “invasive carp” due to similar egg characteristics among these species. The training data contained other species in the same family as invasive carp (Cyprinidae), so the family of Cyprinidae excluding invasive carp was treated as a separate category in the family level response variable. The distribution of eggs per species was imbalanced with invasive carp, Freshwater Drum (Aplodinotus grunniens), and Emerald Shiner (Notropis atherinoides) comprising most of the eggs in the training data (Table 2).

Table 2 Training data taxonomic levels.

The table includes the taxonomic levels and number of eggs per species in the training data collected from pools 14–20 of the Upper Mississippi River during 2014–2016. The eggs with a label of “species unidentified” were eggs where the genetic analysis was able to identify a genus but not a species.

Family	Genus	Common name (species)	Number of eggs in training data	
Catostomidae	Carpiodes	Carpsuckers species unidentified	1	
Quillback (cyprinus)	1	
River Carpsucker (carpio)	8	
Ictiobus	Bigmouth Buffalo (cyprinellus)	7	
Black Buffalo (niger)	1	
Buffalo species unidentified	10	
Smallmouth Buffalo (bubalus)	2	
Clupeidae	Alosa	Skipjack Shad (chrysochloris)	1	
Dorosoma	Gizzard Shad (cepedianum)	2	
Cyprinidae	Cyprinella	Spotfin Shiner (spiloptera)	6	
Luxilus	Common Shiner (cornutus)	1	
Macrhybopsis	Silver Chub (storeriana)	36	
Speckled Chub (aestivalis)	28	
Notropis	Channel Shiner (wickliffi)	32	
Emerald Shiner (atherinoides)	201	
River Shiner (blennius)	16	
Sand Shiner (stramineus)	1	
Shiner species unidentified	69	
Pimephales	Fathead Minnow (promelas)	5	
Hiodontidae	Hiodon	Goldeye (alosoides)	7	
Invasive Carp	Invasive Carp	Invasive Carp	782	
Moronidae	Morone	Striped Bass (saxatilis)	17	
White Bass (chrysops)	1	
Percidae	Etheostoma	Banded Darter (zonale)	1	
Percina	Common Logperch (caprodes)	1	
Sander	Walleye (vitreus)	2	
Sciaenidae	Aplodinotus	Freshwater Drum (grunniens)	733	

We trained the WhoseEgg random forests using the randomForest R package (Liaw & Wiener, 2002). Each model was trained with 1,000 trees. The other tuning parameters were set to the default values in randomForest. Parameters were specified to be consistent with Camacho et al. (2019) and Goode et al. (In press). WhoseEgg returns several values from the random forests for each egg observation in the uploaded data: Random forest probabilities: Random forests are ensembles of many trees (1,000 in the case of WhoseEgg), and each tree returns a prediction. The proportion of trees that return a prediction of a particular level (on out-of-bag observations) is interpreted as the probability that a randomly generated tree (under the conditions used by a random forest) will predict an observation to be in a specific level (Cutler et al., 2007). WhoseEgg returns the random forest probability for each level within the family, genus, and species levels contained in the training data.

Random forest prediction: The response variable level with the highest random forest probability is considered the random forest prediction. WhoseEgg returns the random forest prediction for the family, genus, and species levels.

Egg characteristic collection

WhoseEgg requires users to collect and provide 15 variables associated with an egg: the year and day of egg collection and all variables listed in Table 1 except for Julian day, embryo diameter coefficient of variation, membrane diameter coefficient of variation, and the embryo to membrane ratio. WhoseEgg internally computes the four excluded variables from the provided variables.

The WhoseEgg ‘Help’ page contains detailed descriptions of these variables to assist the user in the data collection process. For each egg characteristic, the page provides a definition of the variable including units, the name of the variable and required format for the WhoseEgg spreadsheet, and whether the variable is required for upload or computed after upload (Fig. 5A). Additional information is provided with some of the variables. For example, the range of the variable in the training data is provided for continuous variables to allow users to determine if their observations fall within the training data range, and example photos are provided to help with measurements that require additional measurement details (Fig. 5B).

Figure 5 Example ‘Help’ page egg characteristic information.

(A) The WhoseEgg ‘Help’ page provides detailed information about the required variables such as conductivity. (B) Some of the variables such as larval length contain additional information and figures that aid in describing how the variable should be measured.

The environmental variables of temperature and conductivity should be collected at the time of egg collection. The egg morphological variables are best measured in a laboratory. The morphological variables in the WhoseEgg training data were collected by first placing an egg in a petri dish with just enough ethanol to cover the egg and help hold it stationary. Then a photograph was taken of the egg using an Olympus SXZ7 microscope (Image Pro 7.0 software; Media Cybernetics, Bethesda, MD, USA) at two times magnification. Camacho et al. (2019) states that, “For eggs with an embryo, the pictures were taken in the dorsal, ventral, and lateral positions in relation to the embryo. If an embryo was not identifiable, a picture was taken after a quarter rotation of the egg on its y-axis, x-axis, and again on its y-axis.” The quantitative measurements (e.g., larval length and membrane diameter average) were obtained using Image Pro software. The qualitative measurements (e.g., pigment presence and egg stage) were determined visually based on the criteria described on the WhoseEgg ‘Help’ page. Users with questions about variable collection are encouraged to reach out to the app developers for clarifications.

Example

Here, we present an example using WhoseEgg to obtain predictions on a set of fish eggs from the WhoseEgg training data collected in 2016 at the mouth of the Iowa River in Pool 18 of the Upper Mississippi River. This location was selected since it is an area that has been actively monitored to observe invasive carp reproduction along the invasion front (Camacho et al., In press). The data set contains the egg characteristics measured on 215 fish eggs.

Data Input Page. We first upload the spreadsheet ‘example-data.csv’ (available in the Supplemental Material) containing the egg characteristics (gold short-dashed box in Fig. 2). The spreadsheet has the same format as the downloadable template (red solid box in Fig. 2) except that we added variables for site and river of collection. If the uploaded spreadsheet has no formatting errors, WhoseEgg prints two interactive tables under ‘Egg Characteristics’ (blue long-dashed box in Fig. 2). The table in the ‘Input Data’ tab contains the uploaded data (Fig. 2) and the table in the ‘Processed Data’ tab contains a dataset created by WhoseEgg with the predictor variables. Note that the variables of Year and Day are included in the input data table but will not be in the processed data table since they are not used as predictor variables by the models. The variable of Julian Day, however, will be computed based on these variables and added to the processed data. The user can filter and sort the data in the tables as desired.

Predictions Page. After uploading the data, we navigate to the ‘Predictions’ page. We click on the ‘Get Predictions’ button (solid red box in Fig. 6) and a ‘Table of Predictions’ (gold short-dashed box in Fig. 6) and ‘Visualizations of Predictions’ (blue long-dashed box in Fig. 6) appear. The table contains seven variables. The first variable is the Egg ID provided in the uploaded spreadsheet. The remainder of the variables are the random forest predictions (variables ending with ‘Pred’) and corresponding random forest probabilities (variables ending with ‘Prob’) for the family, genus, and species.

Figure 6 ‘Predictions’ page.

After the ‘Get Predictions’ button was clicked for the example, the ‘Predictions’ page displays the depicted content.

The plots on the ‘Summary of Predictions’ tab summarize the predictions of all observations in the uploaded data. A bar chart is created for each taxonomic level showing the levels included in the random forest predictions and the number of predictions per level (blue long-dashed box in Fig. 6). In our example data, most predictions fall in the family of Sciaenidae, the genus of Aplodinotus, and the species of Freshwater Drum. The second most frequent category in each taxonomic level is invasive carp. Note the number of invasive carp predictions varies from 55 at both the family and genus levels to 57 at the species level. Additional work outside of WhoseEgg could be done after the predictions are downloaded to investigate which observations are predicted differently across the taxonomies by the random forests. This could provide insight into why the random forests made different predictions for these observations.

The ‘Individual Egg Predictions’ tab allows users to select an egg of interest by clicking on a row in the ‘Table of Predictions’ (Fig. 7A). Then bar charts are created showing the random forest probabilities corresponding to the selected egg for all categories within family, genus, and species (Fig. 7B). In our example, 56 eggs are predicted to have a species of invasive carp. After the predictions were downloaded, it was found that 47 (84%) of the eggs have a random forest probability greater than 80% for invasive carp. While the model mostly returns invasive carp predictions with high random forest probabilities, it may be of interest to further explore the eggs with lower random forest probabilities. Here, we consider the egg with the lowest random forest probability for invasive carp out of the eggs with a random forest species prediction of invasive carp (Fig. 7A; egg 77). Only 37% of the trees voted for egg 77 to be an invasive carp. Since this is a low percentage, we are interested in knowing what other species received votes from the random forest trees. The bar chart of random forest species probabilities for egg 77 shows that approximately 27% of the trees in the species random forest voted for Speckled Chub (Macrhybopsis aestivalis) and approximately 17% of trees in the species random forest voted for Freshwater Drum.

Figure 7 ‘Individual Egg Predictions’ visualizations.

(A) The ‘Table of Predictions’ from the example data was filtered to only include rows with at least one prediction of invasive carp and sorted from lowest to highest species probability. The egg with the lowest random forest probability of invasive carp (the first row) was selected. (B) The ‘Individual Egg Predictions’ tab then generates bar charts of the random forest probabilities for all categories within a taxonomic level corresponding to the egg selected.

Downloads Page. With the predictions obtained, we move to the ‘Downloads’ page. We first click on the ‘Preview Data’ button (solid red box in Fig. 8) to preview the spreadsheet available for download. The table is too long to show all the columns at once, but a horizontal scrolling option allows the user to see all columns. The table includes all initial variables uploaded to WhoseEgg, additional predictor variables computed by WhoseEgg, and random forest predictions and probabilities for all categories within the three taxonomic levels. We then select a file type (xlxs, xls, or csv) for download (gold short-dashed box in Fig. 9) and click the ‘Download Predictions’ button (blue long-dashed box in Fig. 9).

Figure 8 ‘Downloads’ page.

The image depicts the content displayed on the ‘Downloads’ page for the example after the ‘Preview Data’ button was clicked.

Figure 9 Violin plots of species random forest probabilities for invasive carp.

The plots are separated by the species random forest predictions.

After Download. The downloaded results may be used for further investigation. For example, we explore the relationship between the invasive carp random forest probabilities and predictions for the species model. We create a violin plot of the species predictions vs. the random forest probabilities that an egg is an invasive carp (Fig. 9). Most of the eggs predicted to be invasive carp have high probabilities (above 0.8) of being invasive carp and most of the eggs predicted to be a species other than invasive carp have low invasive carp random forest probabilities (below 0.25). We could elect to genetically identify a handful of eggs predicted to be invasive carp with low probabilities. This would enable us to gain confidence in those eggs. The results from WhoseEgg suggest there are invasive carp eggs present at the mouth of the Iowa River and help us to identify a possible subset of eggs for genetic identification if deemed necessary.

Limitations and user responsibility

WhoseEgg is a powerful tool for classifying fish eggs without the need to obtain costly genetic identifications, but as with all models, there are assumptions and limitations users must be aware of. First, the random forests used by WhoseEgg at the time of writing this article were trained and validated on data from the Upper Mississippi River basin. Due to possible variation in fish egg characteristics across regions, additional validation is required to know how the models will perform in other regions. One option for users in other regions is to perform their own validation similar to the one in Goode et al. (In press) by first applying WhoseEgg to genetically identified eggs from the region of interest. If the models perform well on invasive carp, it provides evidence that WhoseEgg will return trustworthy predictions on future eggs from that region. If the models do not perform well, new random forest models could be developed using similar approaches as Camacho et al. (2019) and Goode et al. (In press) to identify fish eggs in different regions or across multiple regions.

A second limitation of WhoseEgg is that models are only able to return predictions that are in the training data taxonomic levels (Table 1). If a new collection of eggs contains a different family, genus, or species, WhoseEgg will not be able to correctly predict the egg. If other species are likely be present in egg collections, WhoseEgg should be used with caution. If the other species have characteristics that vary from invasive carp, WhoseEgg could still be a useful tool for identifying invasive carp. However, if the other species have similar characteristics to invasive carp, WhoseEgg may incorrectly predict these eggs as invasive carp.

A third limitation is that the validation of the random forest in WhoseEgg was focused on the classification of invasive carp and not other species present in the training data. Random forests generally were successful at predicting the identity of other fish eggs, but because the success of identifying other species was not specifically assessed, we urge users to be cautious if there is an interest in focusing on the identification of different species. As with the regional limitation, users who are interested in applying WhoseEgg to identify other species could perform a validation focusing on the other species of interest. If the WhoseEgg training data contain a large amount of the species of interest, the validation could be performed on the WhoseEgg training data. Otherwise, a new dataset with more observations from the species of interest should be used.

The three limitations discussed indicate that a user of WhoseEgg has the responsibility to acknowledge if their data are not appropriate to use with the WhoseEgg models. In addition to considering the location of data collection and possible fish species present in the data, users should also consider whether the egg characteristics in their data fall in range of the training data egg characteristics (see the app ‘Help’ page). If egg characteristics fall outside of the training data ranges, the random forests will be forced to extrapolate, which could lead to untrustworthy predictions. WhoseEgg alerts users if the uploaded egg characteristics fall outside of the training data values, but the final check of data correctness is the responsibility of the user.

Future work

There are many possibilities for updates to WhoseEgg. As additional eggs are collected, the random forests within WhoseEgg can be updated. This could include adding eggs from other species (e.g., Black Carp; Mylopharyngodon piceus) and regions (e.g., Ohio River Valley) not currently included in the training data. Models for predicting species other than invasive carp (e.g., Walleye; Sander vitreus) could also be trained. In regards to model extrapolation outside of the training data values, WhoseEgg already returns warning messages if an observation falls outside of the range of one variable. However, the procedure is done independently for each variable, which ignores correlation between variables. A procedure could be developed and implemented to determine if an observation falls within the joint range of multiple variables. Another possibility to improve the random forest ability to predict well on unseen data is feature selection. Camacho et al. (2019) explored models with a reduced set of input variables, which performed well, but these models were not implemented in WhoseEgg. Furthermore, additional resources for app usability could be developed (e.g., video demonstrations of variable measurements and app use).

WhoseEgg also provides inspiration for the development of other tools for egg classification. For example, future work could explore the use of convolutional neural networks to classify egg species given an image of an egg. This would provide a more streamlined approach for classification that removes the process of taking manual measurements of morphological variables. A convolutional neural network could be incorporated in a phone app that allows users to take a picture of a fish egg with a phone camera and return a prediction. Such a tool could possibly be used in the field for immediate data-based evidence of the classification of the egg.

Beyond the identification of fish eggs, this web-based application demonstrates the potential for fisheries scientists to make their work more accessible to other professions in our field. We encourage others to develop similar web-based tools for other complicated models that will make them more accessible and help to facilitate their use and application.

Supplemental Information

Supplemental Information 1 Template provided by WhoseEgg for inputting egg data.

The spreadsheet contains “helpers” to assist users correctly format the data.

Click here for additional data file.

Supplemental Information 2 Code use to prepare data for WhoseEgg (R markdown version).

Click here for additional data file.

Supplemental Information 3 Code use to prepare data for WhoseEgg (PDF version).

A PDF generated from the ‘preparing-data-for-app.Rmd’ file with the code used prepare the egg data for the WhoseEgg app along with descriptions and figures.

Click here for additional data file.

Supplemental Information 4 Example egg data used in the article.

This example data contains eggs characteristics on a set of 215 fish eggs collected in 2016 at the mouth of the Iowa River in Pool 18 of the Upper Mississippi River.

Click here for additional data file.

We thank C. Camacho and A. Matthews for providing data and additional information about previous work.

Additional Information and Declarations

Competing Interests

Author Contributions

Animal Ethics

Field Study Permissions

Data Availability

The authors declare that they have no competing interests.

Katherine Goode conceived and designed the experiments, performed the experiments, analyzed the data, prepared figures and/or tables, authored or reviewed drafts of the article, developed code, and approved the final draft.

Michael J. Weber conceived and designed the experiments, analyzed the data, authored or reviewed drafts of the article, and approved the final draft.

Philip M. Dixon conceived and designed the experiments, authored or reviewed drafts of the article, and approved the final draft.

The following information was supplied relating to ethical approvals (i.e., approving body and any reference numbers):

The data used in this article were collected under approval from the Iowa State University Institutional Animal Care and Use Committee Protocol 7-13-7599-I.

The following information was supplied relating to field study approvals (i.e., approving body and any reference numbers):

Iowa DNR gave permission for field sampling (SC1037).

The following information was supplied regarding data availability:

The Excel template provided by WhoseEgg for users, the code used to prepare the egg data used by WhoseEgg, and the example dataset used in the article to demonstrate how WhoseEgg works are available in the Supplemental Files.

These materials and additional materials (i.e., the code, random forests, and training data associated with WhoseEgg) are available in the WhoseEgg GitLab repository (https://github.com/goodekat/WhoseEgg) and in the Iowa State University digital repository: https://doi.org/10.25380/iastate.15046578.v1.

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
