# Peer review of "WhoseEgg: classification software for invasive carp eggs"

_PeerJ, doi:10.7717/peerj.14787_

## Round 0.1 · original submission · Minor Revisions

Both reviewers appreciated the work you've done with this paper, and have suggested a couple of minor revisions that I think will improve the manuscript.

·

Basic reporting

A few minor suggestions in the attached document.

Experimental design

no comment

Validity of the findings

no comment

Additional comments

A few minor suggestions in the attached document.

Reviewer 2 ·

Basic reporting

Overall the paper is well-written and the Shiny app is presented well. A few comments:

1. I do wonder whether all of the figures are needed (in particular Fig 4, whether both of Figs 6 and 7 are needed) as many of them are screenshots of the Shiny app. It's possible some could be replaced with more informative figures about some of the egg characteristics used in the random forest modeling itself.

2. I wasn't convinced the Future Work section, as written, added much to the paper as it seemed mostly speculative. It could be largely removed, as much of it is addressing issues already discussed under Limitations, or the authors should work to expand and make it less speculative, giving more concrete ideas and methods for future use of the application.

Experimental design

The method is interesting and is clearly of use under the right circumstances. My largest concern is that the authors leave much of the actual methodology out of the paper, instead citing previous works. While I think that's OK for some of the random forest details, if the idea of this paper is to get people ready to use WhoseEgg, more time should probably be spent on the variables that go into the modeling. I see that there is a long section describing all the measurements in the WhoseEgg app, but this wasn't mentioned in the paper. In addition, it wasn't clear to me how consistently some of the variables might be measured/identified across different researchers, especially some of the developmental and coloration characteristics. How much might user error influence some of the predictions? How much training should someone have to be able to confidently identify egg characteristics and use this app? So, more methods about what needs to be measured and how could allow researchers to understand the underlying knowledge base that might be needed to use WhoseEgg.

Validity of the findings

I went on the app and ran the example data and replicated the results shown in the paper, so it seems to be in working order. I think the limitations discussed by the authors are correctly identified and it possible even more caution should be recommended when extrapolating outside of the region/species training datasets, because as of now that is the strongest limiting factor on WhoseEgg's use.

Additional comments

I think many of my concerns are described above, but also noted by the authors themselves in the Limitations section. This app is interesting, but it does seem like its use might be more or less limited to the Upper Mississippi basin and the identification of invasive carp eggs among a relatively limited set of other species. Perhaps the Future Work section could be better spent thinking of how the app might be updated and/or remain a living app into the future, so as to increase its quality across a geographic or taxonomic scope. Otherwise this risks becoming an app with a relatively narrow use-case.

---

## Round 0.2 · accepted · Accept

Thanks for addressing the previous reviewers' comments, the new version looks significantly improved!